# Association between Dentofacial Features and Bullying from Childhood to Adulthood: A Systematic Review

**DOI:** 10.3390/children10060934

**Published:** 2023-05-25

**Authors:** Alice Broutin, Isabelle Blanchet, Thibault Canceill, Emmanuelle Noirrit-Esclassan

**Affiliations:** 1Paediatric Dentistry, University Toulouse III, CHU Toulouse, Centre for Anthropobiology & Genomics of Toulouse (CAGT) CNRS UMR 5288, 31400 Toulouse, France; 2Paediatric Dentistry, UMR 7268 ADES CNRS EFS Aix-Marseille University, 13385 Marseille, France; 3InCOMM (Intestine ClinicOmics Microbiota & Metabolism), UMR 1297, INSERM, 31400 Toulouse, France; thibault.canceill@univ-tlse3.fr; 4Paediatric Dentistry, University Toulouse III, CHU Toulouse, UMR 1297 INSERM, 31400 Toulouse, France

**Keywords:** malocclusion, dental structural defect, dentofacial features, bullying

## Abstract

Bullying occurs when an individual is repeatedly victimised by negative actions performed by peers. As oral features, like malocclusion and dental structural defects, can promote psychological distress, which is also found in those who are bullied, we aimed to study the association between orofacial conditions and bullying. A systematic review (PROSPERO CRD42022331693), including articles dealing with bullying and dentofacial traits, was performed following the PRISMA chart. The iterative search of eligible publications was carried out on 27 March 2023 on four databases (PubMed, PubPsych, Web of Science and Cochrane Reviews) and in the grey literature. Among the 25 articles included, 4 referred to qualitative studies, which analysed 632 interviews with children, 8 interviews with parents, 292 letters, and 321 Twitter posts. The other 21 were cross-sectional studies, which included 10,026 patients from 7 to 61 years old. Two of the qualitative studies and seven of the cross-sectional studies rated a low risk of bias, according to Joanna Briggs Institute’s Critical Appraisal Tools. The majority of studies (88%) reported a relationship between malocclusion or structural defects and exposure to bullying among young adolescents. Structural dental abnormalities and severe malocclusion should be managed, among others, for psychological questions because they crystallise the loss of self-confidence and increase the risk of bullying.

## 1. Introduction

Bullying occurs when a child or adolescent is victimised repeatedly over time by negative actions performed by one or more peers [1]. This aggressive behaviour can manifest itself in different forms, sometimes in combination (verbal, physical, via social media…). Victims tend to be anxious and insecure with low self-esteem. They also have depressive tendencies that persist into adolescence and early adulthood, even after victimisation has stopped [2].

Facial aesthetics serve a great purpose in social interactions [3], and a positive relationship exists between facial attractiveness and interpersonal popularity, as well as others’ favourable evaluation of one’s personality and social behaviour [4].

Dental features such as occlusion, colour or shape of teeth have a significant impact on facial appearance and, more globally, on the general perception of body image. As the eyes are focused on the face in normal social interactions, it is invariably impossible to hide or disguise them [5]. Therefore, if these features deviate from the norm, the general aesthetics of the person concerned may be altered.

Among these deviations, malocclusion is one of the most common oral disorders and can be defined as a significant discrepancy from the ideal occlusion involving a condition of imbalance in the position of the teeth, facial bones and soft tissues [6]. Even if it is not considered a disease, orthodontic treatments are proven to be necessary to treat both functional and aesthetic alterations and prevent traumatic risks [7,8,9].

From a purely dental perspective, structural defects can be defined as disturbances in dental hard tissue matrices and their mineralisation during the period of odontogenesis [10]. It could include, among others, developmental defects of enamel, dental enamel hypoplasia, amelogenesis or dentinogenesis imperfecta. These last two are rare, hereditary, developmental disorders [11], whereas the others can have an environmental origin. They can affect the structure, clinical appearance and sensitivity of the teeth of both dentitions. Because of aesthetic and masticatory function alterations, guidelines for restorative treatment recommend covering the surface with direct composite resin or composite resin veneers in young children and adolescents until adulthood [12].

Abnormal dental features like malocclusion and dental structural defects are increasingly visible in early adolescence, with the eruption of permanent teeth and maxillomandibular growth. These orofacial features can lead to significant anxiety and other emotional or behavioural problems [1,10]. Their treatment usually occurs at a crucial stage of psychosocial development, which is also a period of high exposure to bullying.

Given that malocclusion and dental defects can promote psychological distress and decrease self-esteem [13,14], which are also personality traits found in bullied [15], we aimed to study the association between these orofacial conditions and bullying. We hypothesised that patients with malocclusions or teeth anomalies were prone to bullying.

## 2. Material and Method

### 2.1. Study Design

This study has been conducted as a systematic review of the literature. The protocol has been established in accordance with the recently updated PRISMA grid [16] and registered on the International Prospective Register of Systematic Reviews (PROSPERO) database under the number CRD42022331693. The Joanna Briggs Institute (JBI) Manual has been used to evaluate bias [17].

### 2.2. Eligibility Criteria

We included all types of studies, quantitative and qualitative, assessing both bullying and dental traits such as malocclusion and tooth abnormality or dental defects. The samples had to include children, adolescents or their parents. Studies published in English, French, and/or Spanish were eligible to be included. We excluded case reports, literature reviews, and studies looking at orofacial clefts, oligodontia or patients with dental caries.

### 2.3. Sources of Information and Search Strategy

The iterative search of eligible publications was carried out on 27 March 2023 on four databases (PubMed, PubPsych, Web of Science and Cochrane Reviews) and in the grey literature. The search procedure submitted was: (tooth abnormality OR malocclusion OR amelogenesis imperfecta OR dentinogenesis imperfecta OR dental enamel hypoplasia OR developmental defects of enamel) AND (bullying OR relational aggression OR cyberbullying OR emotional abuse OR harassment OR school violence OR teasing OR victimization) NOT (carie) NOT (decay).

The reference list of each study selected has been analysed to identify additional studies that had not been found in the first search.

### 2.4. Study Selection

Results obtained from the database research were exported to Rayyan^®^ software [18]. Duplicates have been detected and removed. Two evaluators (AB and EN) independently reviewed the titles and abstracts of the publications in order to determine whether or not they met the eligibility criteria mentioned above. When in doubt, the full text was analysed to determine if the article was suitable for inclusion. If the reviewers were not blinded by authorship or the name of the journals, they were blinded to each others’ decisions.

Disagreements between individual judgements were planned to be resolved by asking a third reviewer (TC) for their opinion.

### 2.5. Data Collection

The full-text review of selected articles enables systematic extraction of the following data: authorship, year and country in which the study was performed, type of study, main objective, information on the study sample features (number, age, inclusion and exclusion criteria), how dental characteristics and bullying exposure were assessed, the tools used for data collection, the tests used for statistical analyses and the main outcomes.

### 2.6. Bias Assessment

In order to assess the trustworthiness, relevance and results of the selected articles, Checklists for Analytical Cross-Sectional Studies and for Qualitative Research from Joanna Briggs Institute’s (JBI) Critical Appraisal Tools were used [17].

For the qualitative research assessment, each question in the checklists could be answered as “yes”, “no”, “unclear” or “not applicable”. A “yes” response was quantified by 1 point, a “no” response by 0 points and an “unclear” response by 0.5 points. The total score was then reported out of 100. Two evaluators (AB and EN) independently assessed each domain of the checklists.

Concerning the cross-sectional studies evaluation, question 3 received a “no” response when dentofacial condition (dental features, malocclusion and/or dental structure abnormalities) was assessed using a self-reported questionnaire but not a validated tool. In the same way, question 7 received a “no” response when bullying was assessed in a non-validated way. Questions 5 and 6 were answered as “no” when no confounding factors had been identified, and no strategy to deal with them had been developed. Question 8 was answered as “no” when adequate statistical tests have not been used considering variables.

The risk of bias would be rated as high when the study reached a score of less than 49%, moderate when it reached a score between 50% and 69%, and low when it reached a score of more than 70% [19].

## 3. Results

### 3.1. Study Selection

A total of 171 references were obtained after research was conducted on four databases (PubMed, PubPsych, Web of Science, Cochrane Reviews) and grey literature. After the removal of duplicates, 147 articles were screened by a careful reading of their titles and abstracts. A further 120 of them were then excluded because they did not correspond with the aim of this review leaving 27 articles selected for full-text reading and assessment of the eligibility criteria. After full-text reading, two more articles were excluded: one did not focus on the subject of this research, and the other was a case report. In the end, 25 articles were included fulfilling the eligibility criteria (Figure 1).

### 3.2. Characteristics of the Selected Studies

Selected articles were published from 1980 to 2022. We included studies dealing with adults. One study [20] included adults between 28 and 34 years old and asked them about their exposure to bullying during their schooling. Their malocclusions had been previously assessed during their adolescence (between 13 and 19 years old). Two studies [8,21] included patients with no age limit (both children and adults). We chose to keep these studies because of the presence of children. Three studies [22,23,24] included parents and questioning them about their children’s dental features and exposure to bullying. In total, 22 of the 25 included studies referred only to young people (7–18 years old) and their parents testifying about their child’s experience.

Four of the articles referred to qualitative studies. The other 21 were cross-sectional studies. Three of the qualitative studies analysed interviews with a total of 632 children [25,26] and 8 interviews with parents [23]. Patients included were children and parents of children aged from 9 to 18 years. Two studies analysed written texts: 292 letters (143 from caregivers and 149 from children between 11 and 18 years old) [25] and 321 Twitter posts [27].

Altogether, the 21 cross-sectional studies included 10,026 patients. Two of the studies [28,29] dealt with the same cohort, and one of them [30] included some patients from a previous study [31]. Patients included were aged from 7 to 61 years. Three studies included adults [8,20,21], and two included parents of children under 18 years old [22,24]. Fourteen of these studies included children at school. Six of them included patients or parents from dental, maxillo-facial or orthodontic follow-up, and one of them included parents from social networks.

In order to assess bullying, five studies used the same modified questionnaire from Shaw et al. [26] by Al Bitar et al., in 2013 [22,28,29,32,33], two used the Olweus Bullying/Victim Questionnaire [30,31]; Ramos et al. [9] used the questionnaire used in the National Survey of School Health (PeNSE), Gatto et al. [7] used the Kidscape Questionnaire, Duarte Rodrigues et al. [34] used a question from the CPQ8-10, Bazan-Serrano et al. [35] used a questionnaire from Oliveros et al. [36], Rwakatema et al. [37] used a questionnaire from Ng’ang’a et al. [38] and Onyeaso et al. [39] used a modified questionnaire from Helm [20]. Seven of the studies used individual arrangement questionnaires [20,21,24,40,41,42,43], and Alanko et al. [8] used a structured diary.

To assess malocclusion, eleven of the cross-sectional studies assessed dentofacial features, using a modified self-reported questionnaire from Shaw et al. [26] for five of them [22,28,29,32,35], a self-administered questionnaire from Ng’ang’a et al. [38] for Rwakatema et al. [37], and author-created questionnaires for the last 5 [20,24,40,41,43].

Eight of the selected articles mentioned a practitioner’s assessment of the malocclusion, using the Dental Aesthetic Index (DAI) for five of them [7,9,33,34,39], the Aesthetic Component (AC) of the Orthodontic Treatment Need Index (IOTN) for four of them [8,9,30,31], associated with the Dental Health Component (DHC) of the IOTN for two of them [30,31] or with a patient’s self-perceived need for orthodontic treatment assessment using a modified IOTN-AC scale [8].

Two articles referred to dental structure abnormalities using the modified Developmental Defects of Enamel (DDE) index to assess DDE [34,42] and the modified Dean index to assess dental fluorosis [34].

### 3.3. Identification of Bias in Studies

Only two studies fulfilled all the criteria from the JBI Critical Appraisal Tools checklists [23,34]. Two of the qualitative studies [23,26] and seven of the cross-sectional studies [7,8,29,30,31,34,40] rated a low risk of bias (Table 1 and Table 2).

### 3.4. Results of Individual Studies

Of the 25 articles included, 5 focused only on malocclusion, 3 only on dental structural defects, and 20 more globally on dentofacial features. A total of 22 concluded that there was an association between bullying and oral condition. The percentage of people who were bullied varied greatly from study to study. The results of the four qualitative studies are detailed in Table 3. Those of the 21 cross-sectional studies are detailed in Table 4.

## 4. Discussion

The present literature review showed that the vast majority of studies found a relationship between dental malocclusion or structural defects and exposure to bullying, which occurred in 5.7 to 100% of the samples. Only three studies [9,33,39] did not report an impact of malocclusion on bullying.

Most of the studies included young adolescents. Indeed, the prevalence of bullying seems to be higher in children of 12–13 years and younger [28,29,31], and bullying tends to decrease with increasing age [44]. In order to standardise the source of information across all ages and to enable comparisons between the youngest schoolchildren and the older ones, some authors chose to use parents’ questionnaires [22].

In this literature review, there is great variability in the clinical assessment tools for dental defects. In order to evaluate dentofacial condition, eight studies used clinical examination [23,24] or professional assessment [7,30,31,33,34,42], twelve used patients’ self-evaluation [20,22,25,26,27,28,29,32,35,37,40,43], and three used both [8,9,39]. The originality of our review, and maybe one of its limits, was to take an interest in both malocclusion and structural defects and to regroup all of them under one term: dentofacial features. Considering this, patients’ self-evaluation could be considered a global smile evaluation and not only an evaluation of one precise element of the mouth.

There is no standard instrument to identify and assess bullying [45]. Twenty of the included studies chose to use questionnaires. Whatever the Likert scale, results were divided into “bullied” or “non bullied”. Some authors mentioned that because of the sensitivity of the topics, anonymous questionnaires should enable avoiding the embarrassment of direct interview confrontation [22,28,29] and then enable more sincere responses. As this is the most current approach seen in the literature, we could suppose that previously published questionnaires have been chosen because of their validity and to be able to compare results with those of different other studies [22]. However, 13 of the cross-sectional studies used 8 different validated questionnaires, and the other 7 used author-created questionnaires. This drastically limits the possibility of comparing results.

Even if most of the included articles chose the same type of tool to assess bullying, comparison of results remains difficult also because time intervals of the exposure and cut-offs could be very different between all the studies. Effectively, the time intervals assessed extended from two days [8] to throughout schooling [20,37]. Most of the studies assessed exposure to bullying during one [22,28,29,33,34] or two months [30,31,40], probably to limit the risk of bias related to long-term memory. However, this adds the risk of underestimating the exposure to bullying of certain subjects. With regard to the cut-off, for example, during a period of one month, subjects were considered as victims of bullying from one [34] to two episodes of teasing [31,40], or if episodes were qualified as “always or almost always” for Ramos [9]. This emphasises the absence of consensus regarding the frequency of abuse.

This review highlights the fact that all authors are not interested in the same events: some of them assess the frequency of ‘nicknames’ or ‘name-calling’ [28,29,34], while others evaluate exposure to bullying in different forms: physical [7,22,35], verbal [7,22,34,35], cyberbullying [22,35], emotional, racist or sexual [7]. Moreover, bullying in multiple forms seems to be an emerging issue that has never been addressed, and that warrants attention [22]. Terminology could be crucial: according to Ross [46], teasing should not always be identified as bullying, but as a form of acceptance and dialogue among friends, with no significant harm intended to the recipient. However, peer aggression experiences that do not meet the bullying criteria can also be rated as harmful by victims [47]. Nevertheless, as soon as it results in harm and psychological distress, this aggressive behaviour should be considered bullying [31]. Bullying frequently comes from peers in the school setting, but unfortunately can also occur within the family [48].

According to JBI Critical Appraisal Tools, less than half of all the included studies rated a low risk of bias [7,8,23,26,29,30,31,34,40], and four of the cross-sectional ones rated a high risk of bias [21,24,37,43]. Standardisation and refinement of assessment methodologies for these topics would be beneficial to improve research in the area of bullying and dentofacial characteristics.

Comparing results between studies performed in different countries with different socio-cultural contexts could be irrelevant because of differences in anti-bullying policies, the prevalence of dental structural defects or malocclusions and inaccessibility to aesthetic or orthodontic treatments. Twenty-five articles were included. Of the total, eight studies were performed in European countries [8,20,23,26,30,31,40,41], seven in South and Central America Countries [7,9,21,33,34,35,43], three in African countries [32,37,39], three in Middle East countries [22,28,29], one in Asia [42], one in Oceania [25] and one in North America [24]. Moreover, studies were performed during different periods: two between 1985 and 1993 [20,24], four between 2004 and 2008 [37,39,41,42], and the others after 2010. As bullying policies and aesthetic or orthodontic treatments have evolved during these years, it is possible that the results may be different.

Considering exposure in terms of gender, some authors showed a significant difference, with more boys being victims than girls [22,28,35], while others underlined no significant differences [31,32,35]. However, differences could be more subtle: when males are more likely to endure direct forms of aggression, such as physical attacks, in contrast, females are exposed to more indirect types, which could be underestimated [49].

Four of the included articles [22,26,28,40] found that teeth were one of the most frequent physical features targeted in people who were victims of bullying. Only one of them included patients from an orthodontic population, whereas the three others used school and social networks, so these results could be considered representative. The most frequently identified dental features targeted were prominent maxillary anterior teeth [20,22,24,27,28,32,40,49], spaced or missing teeth [22,27,28,32,40] and shape and colour of teeth [22,27,28,32], which can be qualified as conspicuous dentofacial characteristics. In Helm et al., study, 50% of the subjects with the most extreme maxillary overjets had experienced teasing [20].

This review shows that the more the dental condition (malocclusion, enamel defects, dental fluorosis, amelogenesis imperfecta) could be visible because of its severity or because of its anterior situation, the greater the risk of experiencing teasing or bullying [23,34,39,49]. However, in their literature review on the impact of malocclusion, Zhang et al. [4] found that, ironically, milder deviations in tooth position tend to evoke ridicule and teasing, whereas severe deformities will elicit strong emotional reactions such as pity or revulsion.

In their material and methods, most of the studies [9,20,22,28,29,32,33,34,35,39,40] reported that they had excluded individuals with orthodontic appliances to avoid a confounding variable. Due to the conclusions of the study performed by Shaw et al. in 1980 [26], they hypothesised that wearing an orthodontic appliance should increase bullying exposure, whereas the more recent study of Seehra et al. [30] suggested that participants undergoing orthodontic treatment had a significant reduction in bullying. Scheffel et al. [48] underlined the same phenomenon in the short term in patients with dental structural defects after cosmetic treatments. These elements would benefit from being studied in depth in future studies.

A positive consequence of bullying could be the encouragement to initiate orthodontic consultation [27,40] and the motivation to follow the treatments. Therefore, it also influences the expectations regarding orthodontics [40] or cosmetic treatment [12], and communication between the practitioner and the patient is crucial. Effectively, improvement of dental occlusion or aesthetics may not be sufficient to enhance the psychological condition, self-esteem and the patient’s exposure to bullying. In their literature review, Zhang et al. [4] pointed out that after orthodontic treatment, there was little evidence of a marked improvement in the social well-being of the patients. Social network posts examined by Chan et al. [27] confirm the positive psychological impacts of treatments for some victims. Even if it is difficult to determine the longevity and permanency of these positive effects, we know that the negative psychological effects of peer victimisation in school-aged children can continue during the transition to senior school and into adulthood [49]. Then, even if bullying decreases or stops, psychological consequences could persist, which may be devastating to a child, with long-term effects [2,50] leading to an increased rate of suicidal risk and self-injurious behaviour [49,51].

The authors did not limit themselves to the evaluation of the exposure to bullying but also looked at its consequences. Bullying because of dentofacial features increases absenteeism from school [22,28,32], with significantly more bullied students who dislike not only classes but also school outside of classes [32]. Effectively, episodes of bullying frequently take place during break times. Bullying also has negative consequences for academic performance [22]. In Al Bitar et al. study [28], 40% of students believed that bullying harmed their grades.

In three of the studies [7,29,31], bullying was associated with more negative effects of oral condition on Oral Health-Related Quality of Life (OHRQoL). However, it remained unclear whether this negative impact was due to the presence of malocclusion or peer victimisation [31]. We could suppose that the association of these two elements could increase consequences on quality of life.

Studies included in this review reported that individuals not satisfied with their body image [43], with self-hate or low self-esteem [27], or described as introverted [26] had more chances of being victims of bullying. It has already been described by Olweus et al. [2] with tendencies of being anxious and insecure. Some authors have put forward the hypothesis of a ‘victim personality’ [49], which may result from, or be exacerbated by, victimisation [50] and which remains with the individual despite changes in the social situation [49]. It plays a role in the initial instigation of bullying and may be influenced by social background and parenting [50]. This could explain the tendency for children who are victims to remain victims, even when the social situation changes [10], or the occasionally reported phenomenon of being bullied because of having a perfect normocclusion [26,27].

## 5. Conclusions

With regard to targeted physical features, orofacial features are number one. Thus, severe oral conditions like structural dental abnormalities and severe malocclusion should be managed for functional and aesthetic questions, but also psychological ones because they crystallise the loss of self-confidence and increase the risk of harassment. Practitioners who see their young patients during crucial stages of psychological development must be aware of identifying children at potential risk of experiencing bullying, counsel families, and propose early treatments if possible. Explanation about the aetiology of their condition through therapeutic education may also improve their knowledge and help them to cope with negative comments.

## Figures and Tables

**Figure 1 children-10-00934-f001:**
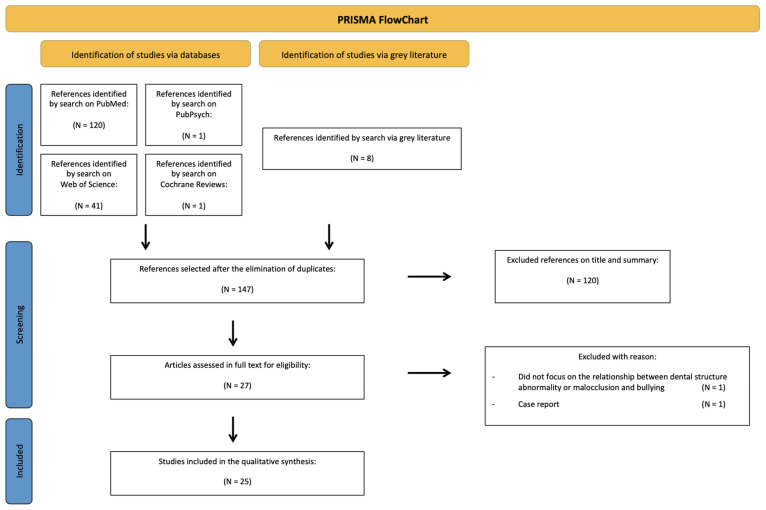
Prisma Flowchart of the research process.

**Table 1 children-10-00934-t001:** Qualitative studies’ risks of bias assessed by checklist for Qualitative Research from Joanna Briggs Institute’s (JBI) Critical Appraisal Tools [17]. The color are related to the bias assessment (green means low risk of biais).

Authors	1. Is There Congruity between the Stated Philosophical Perspective and the Research Methodology?	2. Is There Congruity between the Research Methodology and the Research Question or Objectives?	3. Is There Congruity between the Research Methodology and the Methods Used to Collect Data?	4. Is There Congruity between the Research Methodology and the Representation and Analysis of Data?	5. Is There Congruity between the Research Methodology and the Interpretation of Results?	6. Is There a Statement Locating the Researcher Culturally or Theoretically?	7. Is the Influence of the Researcher on the Research, and Vice-versa, Addressed?	8. Are Participants, and Their Voices, Adequately Represented?	9. Is the Research Ethical According to Current Criteria or for Recent Studies, and Is There Evidence of Ethical Approval by an Appropriate Body?	10. Do the Conclusions Drawn in the Research Report Flow from the Analysis or Interpretation of the Data?	TOT	%
Blanch 2019 [25]	0.5	0	1	1	1	0	0	1	1	1	6.5	65%
Chan 2017 [27]	0	1	0.5	1	1	0	0	0.5	0	1	5	50%
Pousette Lundgren 2019 [23]	1	1	1	1	1	1	1	1	1	1	10	100%
Shaw 1980 [26]	1	1	1	1	1	0.5	1	1	1	1	9.5	95%

(1 = yes—0.5 = unclear—0 = no—NA = not applicable).

**Table 2 children-10-00934-t002:** Cross sectional studies’ risks of bias assessed by checklist for Cross Sectional Studies from Joanna Briggs Institute’s (JBI) Critical Appraisal Tools [17]. The color are related to the bias assessment (green means low risk of biais).

Authors	1. Were the Criteria for Inclusion in the Sample Clearly Defined?	2. Were the Study Subjects and the Setting Described in Detail?	3. Was the Exposure Measured in a Valid and Reliable Way?	4. Were Objective, Standard Criteria Used for Measurement of the Condition?	5. Were Confounding Factors Identified?	6. Were Strategies to Deal with Confounding Factors Stated?	7. Were the Outcomes Measured in a Valid and Reliable Way?	8. Was Appropriate Statistical Analysis Used?	TOTAL	%
Al Bitar 2013 [28]	1	1	0	1	1	0	0	1	5	63%
Al Omari 2014 [29]	1	1	0	1	1	0	1	1	6	75%
Alabdulrazaq 2020 [22]	1	1	0	0	1	0	1	1	5	63%
Alanko 2014 [8]	1	0.5	1	1	1	1	1	0	6.5	81%
Bauss 2023 [40]	1	1	0	1	1	1	1	0	6	75%
Bazan-Serrano 2017 [35]	1	0.5	1	1	0	0	1	1	5.5	69%
Chikaodi 2019 [32]	1	1	0	1	0	0	0	1	4	50%
Duarte Rodrigues 2020 [34]	1	1	1	1	1	1	1	1	8	100%
Fleming 2008 [41]	1	1	0	1	1	1	0	0	5	63%
Gatto 2019 [7]	1	1	1	1	1	0	1	0.5	6.5	81%
Helm 1985 [20]	1	1	1	1	0	0	0	0.5	4.5	56%
Carruitero 2019 [33]	1	0.5	1	1	0	0	1	1	5.5	69%
Kilpeläinen 1993 [24]	1	0.5	0.5	1	0	0	0	0	3	38%
Murillo 2015 [21]	1	0	0	1	0	0	1	0	3	38%
Onyeaso 2005 [39]	1	1	1	1	0	0	0	0	4	50%
Ramos 2022 [9]	1	0.5	1	1	0	0	1	0	4.5	56%
Rwakatema 2006 [37]	1	0.5	0	1	0	0	0	0	2.5	31%
Seehra 2011 [31]	1	1	1	1	1	0	1	0	6	75%
Seehra 2013 [30]	1	0.5	1	1	1	0	1	0.5	6	75%
Sujak 2004 [42]	1	0.5	1	1	1	0	0	1	5.5	69%
Veiga Da Silva Siqueira 2019 [43]	1	0	0	1	0	0	0	1	3	38%

(1 = yes—0.5 = unclear—0 = no—NA = not applicable).

**Table 3 children-10-00934-t003:** Characteristics of qualitative studies included in the systematic review (NS = non specified).

Authorship Year of Publication	Country and Date of Study	Participants			Oral Condition Assessed	Results
		N	Age Range	Inclusion Origin		
Blanch et al., 2019 [25]	New Zealand 2012–2015	292 letters (143 from caregivers, 149 from young people) 18 young people interviews	11–18 years	“Wish for a Smile” New Zealand program	Malocclusion	53% of young people’s letters, 92.3% of treated participants’ interviews and 100% of untreated participants’ interviews talked about bullying and negative comments pre-treatment.
Chan et al., 2017 [27]	No geographic restriction 2010–2014	321 Twitter posts	-	Twitter	Morphological features of teeth (malocclusion, braces and orthodontic appliances, personal attributes or personality traits)	Social media can provide new and valuable information about the causal factors and social issues associated with oral health-related bullying. Importantly, some coping mechanisms may mitigate the negative effects of bullying.
Pousette Lundgren et al., 2019 [23]	Sweden 2015	8 interviews with parents	Parents of children 9–18 years old	Public dental service and paediatric dentistry clinic	Amelogenesis imperfecta	The subtheme “psychosocial stress” included fear of the child being bullied. The findings show that parents of children with severe amelogenesis imperfecta report similar experiences as parents of children with other chronic and rare diseases.
Shaw et al., 1980 [26]	Wales NS	STUDY 1 531 interviews with children Teachers’ questionnaires	9–13 years	Schools	Features targeted in the victims of bullying.	Dental features were the fourth commonest target for teasing. Seven per cent of the total sample reported being teased about their teeth once per week or more. Comments about the teeth appear to be more hurtful than those about other features. Children who were teased specifically about their teeth were twice as likely to suffer harassment than those who were not teased about their teeth.
		STUDY 2 82 children	11–13 years	Schools	Features targeted in the victims of bullying by nickname.	The more deviant the dental arrangement, the more salient will it be.

**Table 4 children-10-00934-t004:** Characteristics of cross-sectional studies included in the systematic review (NS = non specified).

Authorship Year of Publication	Country and Date of Study	Participants			Oral Condition Assessed	Oral Condition Assessment	Bullying Assessment	Additional Characteristics Assessed	Results
		N	Age Range	Inclusion Origin					
Al Bitar et al., 2013 [28]	Jordan 2011–2012	920	11–12 years	Schools	Dentofacial features targeted in the victims of bullying	Structured, anonymous, self-reported questionnaire modified from that of Shaw et al., (1980)	Same questionnaire modified from that of Shaw et al., (1980)	General physical characteristics targeted in the victims of bullying. Feelings toward school and school attendance. Perceived effect on academic performance.	Teeth were the number 1 feature targeted for bullying. The three most commonly reported dentofacial features targeted by bullies: spacing between the teeth or missing teeth, shape or colour of the teeth, and prominent maxillary anterior teeth.
Al Omari et al., 2014 [29]	Jordan 2011–2012	920	11–12 years	Schools	Dentofacial features targeted in the victims of bullying	Structured, anonymous, self-reported questionnaire modified from that of Shaw et al., (1980)	Same questionnaire modified from that of Shaw et al., (1980)	General physical characteristics targeted in the victims of bullying, Feelings toward school and school attendance. Perceived effect on academic performance.	There was a significant relationship between bullying because of dentofacial features and negative effects on oral health–related quality of life.
Alabdulrazaq and Al-Haj Ali 2020 [22]	Saudi Arabia 2019–2020	1028	Parents of children 8–18 years old	Social networks	Dentofacial features targeted in the victims of bullying as reported by parents	Self-reported questionnaire modified from that of Al Bitar et al., (2013)	Same questionnaire was modified from that of Al Bitar et al., (2013).	Sociodemographic profile. Parental opinion about the effect of bullying on their child’s feelings toward school and on school attendance. Bullying’s perceived effect on academic performance. General physical characteristics targeted in the victims of bullying.	With regard to targeted physical features, teeth were the number one target. Tooth shape and colour were the most common dentofacial targets, followed by an anterior open bite and protruded anterior teeth.
Alanko et al., 2014 [8]	Finland NS	89 (60 patients/29 controls)	17–61 years (patients 17–61 years/controls 19–49 years)	Study group: oral and maxillo-facial services/control group: university students	Dental appearance	-Patients’ self-evaluation of dental appearance on a visual analogue scale modified from the Aesthetic Component of the Index of Orthodontic Treatment Need (IOTN-AC). -Professional assessment of patients’ dental appearance with the IOTN-AC.	A structured diary developed by the authors	A modified version of the body image questionnaire (Kiyak 1982), Orthognathic Quality-of-Life. Questionnaire (Cunningham 2000) (OQLQ) Rosenberg self-esteem scale (Rosenberg 1965) (RSES). Acceptance and Action Questionnaire II (Bond 2011) (AAQ II) Symptom Checklist 90 (Derogatis 1973) (SCL-90).	15% of the patients had been bullied. Self-perceived dental appearance was more important to orthognathic quality-of-life and body image than an orthodontist’s assessment.
Bauss and Vassis 2021 [40]	Germany 2015–2019	1020	7–17 years	Orthodontic practices	Dentofacial features targeted in the victims of bullying	Anonymous questionnaires	Anonymous questionnaires	Initiator of treatment, desire for orthodontic treatment, treatment motivation, treatment expectations, and general physical characteristics targeted in the victims of bullying.	Bullied subjects identified teeth and weight as the main targets for bullying. Victims who experienced bullying due to malocclusion initiate orthodontic treatment more often themselves and expect therapy to prevent them from experiencing further bullying.
Bazan-Serrano and Carruitero 2017 [35]	Peru NS	218	11–16 years	Schools	Appearance of teeth/targeted by bullying	Question from Al Bitar et al., questionnaire (2013)	Validated questionnaire (from Oliveros et al.)	-	The frequency of general bullying was 32.57%, and bullying due to dental appearance was 18.81%. General and tooth-related bullying was more frequent among students in public schools.
Chikaodi et al., 2017 [32]	Nigeria 2016	835	12–17 years	Schools	Dentofacial features targeted in the victims of bullying	Structured anonymous self-administered questionnaire modified from that used by Al Bitar et al., (2013)	Same questionnaire was modified from that used by Al Bitar et al., (2013).	-General physical characteristics targeted in the victims of bullying. -Feelings toward school and school attendance. -Perceived effect on grades.	About 43% of respondents reported being victims of bullying, while about 32% had bullied someone else. Bullies frequently targeted general physical and dentofacial appearance.
Duarte Rodrigues et al., 2020 [34]	Brazil NS	390	8–10 years	Schools	-Malocclusion -Dental fluorosis -Developmental Defects of Enamel	-Dental Aesthetics Index (DAI) -Modified Dean index -Modified Developmental Defects of Enamel index	One question from the CPQ-8-10 index	-Untreated caries: DMFT/dmft. -Clinical consequences of untreated caries: PUFA/pufa.	A severe malocclusion, a greater maxillary misalignment and the presence of a tooth with pulp exposure were significantly associated with the occurrence of verbal bullying.
Fleming et al., 2008 [41]	United Kingdom 2003–2004	328	8–17 years or over	Orthodontic department	Appearance of teeth targeted by bullying	Children and parents’ anonymous questionnaires	Children and parents’ anonymous questionnaires	Motivation, understanding and expectation of orthodontic treatment	38% reported teasing related to their dental appearance (of these, only 10% were untroubled by the teasing). Teasing was a commonly reported consequence of malocclusion with a negative psychosocial impact.
Gatto et al., 2019 [7]	Brazil 2014	815	11–16 years	Schools	Malocclusion	DAI	Kidscape questionnaire	-Oral Health related Quality of Life: OHIP-14. -Previous orthodontic treatment. -Desire to fix the teeth to improve one’s appearance.	The need for orthodontic treatment was not associated with the OHRQoL; however, bullying and previous orthodontic treatment had a statistically significant association with this variable.
Helm et al., 1985 [20]	Denmark 1981	758 (maloc-clusion: 606/normoc-clusion: 152)	13–19 years when the occurrence of malocclusion was recorded/28–34 years when question-naires sent	Schools	Malocclusion Dental appearance	Questionnaire	Questionnaire	Orthodontic treatment. Functional disorders. Tooth loss. Body image.	Schoolmates’ teasing occurred seven times more often in the presence of malocclusion.
Carruitero and Julca-Ching 2019 [33]	Peru NS	147	12–18 years	Schools	Need for orthodontic treatment	DAI	Questionnaire from Al Bitar (2013)	-Self esteem: Rosenberg test.	The need for orthodontic treatment in schoolchildren showed no impact on academic performance, self-esteem and bullying. The need for orthodontic treatment did not prove to be a determining factor in the presence of such variables in schoolchildren.
Kilpeläinen et al., 1993 [24]	USA 1989–1990	313	Parents of children under 16 years	Orthodontic clinic	Overjet Alignment	Professional assessment	Self-reported questionnaire	Initiator of treatment, treatment motivation.	44% of the parents reported their child had been teased about the appearance of their teeth. Overjet and misalignment were observed to be significant predictors of the parent’s report of the child being teased.
Murillo et al,. 2015 [21]	Costa Rica NS	18	16–35 years	Faculty of Dentistry of the University of Costa Rica	Amelogenesis Imperfecta	Professional diagnostic	Questionnaire	Emotional aspect and dental treatment.	100% had been teased and had suffered social rejection. Dental professionals need to understand AI not only as defective tooth enamel structures demanding specialist clinical management but also the negative impacts of the condition on the lives of their patients.
Onyeaso and Sanu 2005 [39]	Nigeria NS	614 (malocclusion: 279/normocclusion: 335)	12–18 years	Schools	-Malocclusion -Dental appearance	-DAI -Questionnaire	Questionnaire modified from Helm (1985)	Body image	Teasing had no significant difference between the two groups [without malocclusion vs with malocclusion]. Teasing was mostly reported for the following traits: anterior maxillary irregularity, midline diastema, crowding (maxillary and mandibular segments) and spacing (maxillary and mandibular segments).
Ramos et al., 2022 [9]	Brazil NS	494	12–15 years	Schools	-Self-perceived need for orthodontic treatment -Malocclusion	-IOTN-AC -DAI	Questionnaire used in the National Survey of School Health (PeNSE)	-Socioeconomic conditions -Oral Health-Related Quality of Life (CPQ11–14)	Malocclusion did not correlate with bullying history. However, increased maxillary overjet influences adolescent self-perception, suggesting a potential condition for bullying events.
Rwakatema et al., 2006 [37]	Tanzania NS	298	12–15 years	Schools	Dentofacial features	Self-administered questionnaires from Ng’ang’a et al.	Self-administered questionnaires from Ng’ang’a et al.	-	25.8% of the children reported having been teased due to their malocclusion.
Seehra et al., 2011 [31]	United Kingdom 2007–2008	336	10–14 years	Orthodontic clinics	Orthodontic treatment need	-IOTN-DHC -IOTN-AC	Olweus Bully/Victim Questionnaire	-Self-esteem (Harter’s Self Perception Profile for Children) -Oral Health-Related Quality of Life (CPQ11–14)	The prevalence of bullying was 12.8%. Being bullied was significantly associated with Class II Division 1 incisor relationship, increased overbite, increased overjet and a high need for orthodontic treatment assessed using AC IOTN. Significant relationships also exist between bullying, self-esteem and OHRQoL.
Seehra et al., 2013 [30]	United Kingdom 2010–2011	27	Mean age: 14.6 years, standard deviation 1.5	Orthodontic clinic	Orthodontic treatment need	-IOTN-DHC -IOTN-AC	Olweus Bully/Victim Questionnaire	-Self-esteem (Harter’s Self Perception Profile for Children) -Oral Health-Related Quality of Life (CPQ11–14)	Following the commencement of orthodontic treatment, 21 (78 %) participants reported they were currently no longer being bullied due to the presence of their malocclusion. Orthodontic treatment may have a positive effect on adolescents experiencing bullying related to their malocclusion and their OHRQoL.
Sujak et al., 2004 [42]	Malaysia NS	1024	16 years	Schools	Developmental defects of enamel	Modified Developmental Defects of Enamel Index (FDI, 1992)	Self-administered questionnaire	-	About two-thirds of the sample (67.1%) had at least one tooth affected by enamel defects. Only 5.7% had experienced being teased by their friends about the problem.
Veiga Da Silva Siqueira et al., 2019 [43]	Brazil NS	381	12–15 years	Schools	Self-perception about oral health	Questionnaire	Questionnaire	Self-perception about body image (questionnaire)	The prevalence of bullying was 29.6%. Those who indicated that they were criticised due to the condition of their teeth had a 4.37 times greater chance of victimisation. Those who felt that oral health had little effect on their relationship with other people had a 2.2 times greater chance of suffering from bullying than those who did not. It was possible to observe an association between bullying and dissatisfaction with oral health and body image.

## Data Availability

No new data were created or analyzed in this study. Data sharing is not applicable to this article.

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
