# Peer review of "Association between Dentofacial Features and Bullying from Childhood to Adulthood: A Systematic Review"

_children, 2023, doi:10.3390/children10060934_

Round 1

Reviewer 1 Report

It is an extremely cutting-edge study that connects dentistry to other scientific domains such as sociology and psychology. The study design is suitable. The methods used is correct. The writers extracted data in great detail and explained the results in a thorough manner. The discussion part is well-organized, and it covers the various aspects of the subject. It is a quite interesting paper in my perspective.

Author Response

We are grateful to the reviewer for her/his careful reading of our manuscript and the positive comments.

Reviewer 2 Report

Dear authors. The topic of the study is interesting, and we appreciated the efforts of the authors. there are some eye raising doubts throughout the study.

Introduction 

The introduction needs lot of improvement. We ask the authors to have a continuity between the paragraphs in the introduction. Dental features to malocclusion and from malocclusion to dental structural defects and then linking to bullying and its association. 

a. the initial introduction should start with a paragraph on the bullying and its effect and then link it to malocclusion and dental structural defects.

b. the authors can add studies which have shown the association of these defects to the bullying ..

Materials and methods

a. exclsuion and inclusion criteria to be described properly.

b. Has the study been registered in Prospero. 

discussion

it has been written well

English revision required

Author Response

We are very grateful to the reviewer for her/his careful reading of our manuscript and the thoughtful comments.

We have modified the introduction as requested.

"Bullying occurs when a child or adolescent is victimised repeatedly over time by negative actions performed by one or more peers [1]. This aggressive behaviour can manifest itself in different forms, sometimes in combination (verbal, physical, via social media…). Victims tend to be anxious and insecure with low self-esteem. They also have depressive tendencies that persist into adolescence and early adulthood, even after victimisation has stopped [2].

Facial aesthetic serves great purpose in social interactions [3], and a positive relationship exists between facial attractiveness and interpersonal popularity, as well as others’ favourable evaluation of one’s personality and social behaviour [4].

Dental features such as occlusion, colour or shape of teeth have a significant impact on facial appearance and more globally on general perception of body image. As the eyes are focused on the face in normal social interaction, it is invariably impossible to hide or disguise them [5]. Therefore, if these features deviate from the norm, the general aesthetics of the person concerned may be altered.

Among these deviations, malocclusion is one of the most common oral disorders and can be defined as a significant discrepancy from the ideal occlusion involving a condition of imbalance in the position of the teeth, facial bones and soft tissues [6]. Even if it is not considered as a disease, orthodontic treatments are proven to be necessary to treat both functional and aesthetic alterations and prevent traumatic risks [7–9].

From a purely dental perspective, structural defects can be defined as disturbances in dental hard tissue matrices and their mineralisation during the period of odontogenesis [10]. It could include, among others, developmental defects of enamel, dental enamel hypoplasia, amelogenesis or dentinogenesis imperfecta. These last two are rare hereditary developmental disorders [11], when the others can have an environmental origin. They can affect the structure, clinical appearance and sensitivity of the teeth of both dentitions. Because of aesthetic and masticatory function alterations, guidelines for restorative treatment recommend covering the surface with direct composite resin or composite resin veneers in young children and adolescents until adulthood [12].

Abnormal dental features like malocclusion and dental structural defects are increasingly visible in early adolescence, with eruption of permanent teeth and maxillomandibular growth. These orofacial features can lead to significant anxiety and other emotional or behavioural problems [1,10]. Their treatment usually occurs at a crucial stage of psychosocial development, which is also a period of high exposure to bullying.

Given that malocclusions and dental defects can promote psychological distress and decrease self-esteem [13,14], which are also personality traits found in bullied [15], we aimed to study the association between these orofacial conditions and bullying. We hypothetised that patients with malocclusions or teeth anomalies were prone to bullying."

We have added two references dealing with the personality traits of bullied (Mullan VMR, Golm D, Juhl J, Sajid S, Brandt V. The relationship between peer victimisation, self-esteem, and internalizing symptoms in adolescents: A systematic review and meta-analysis. PLoS One. 2023 Mar 29;18(3):e0282224. doi: 10.1371/journal.pone.0282224. PMID: 36989220; PMCID: PMC10058150) and the decrease self-esteem of people with malocclusion (Atik E, Önde MM, Domnori S, Tutar S, YiÄŸit OC. A comparison of self-esteem and social appearance anxiety levels of individuals with different types of malocclusions. Acta Odontol Scand. 2021 Mar;79(2):89-95. doi: 10.1080/00016357.2020.1788720. Epub 2020 Jul 9. PMID: 32643491, and Akpasa, I.O.; Yemitan, T.A.; Ogunbanjo, B.O.; Oyapero, A. Impact of Severity of Malocclusion and Self-Perceived Smile and Dental Aesthetics on Self-Esteem among Adolescents. J. World Fed. Orthod. 2022, 11, 120–124, doi:10.1016/j.ejwf.2022.05.001.).

We have modified the description of exclusion and inclusion criteria as followed:

We included all types of studies, quantitative as qualitative, assessing both bullying and dental traits such as malocclusion and tooth abnormality or dental defects. The samples had to include children, adolescents or their parents. Studies published in English, French, and/or Spanish were eligible to be included. We excluded case reports, literature reviews, and studies looking at orofacial clefts, oligodontia or patients with dental caries.

The study is registered in Prospero under the ID number CRD42022331693.

Author Response

We thank the reviewer for his/her comments.

As requested by the reviewer, the title has been corrected as followed: Association between dentofacial features and bullying from childhood to adulthood : a systematic review

The term “abnormal” has been removed to remain neutral towards these patients.

The reviewer is completely right and this point (bullied/bullies) is now corrected in the new version of the manuscript: “bullies” has been replaced by “bullied”.

Authors have fully revised references and added new references to provide stronger evidence:

De Ridder, L.; Aleksieva, A.; Willems, G.; Declerck, D.; Cadenas de Llano-Pérula, M. Prevalence of Orthodontic Malocclusions in Healthy Children and Adolescents: A Systematic Review. Int. J. Environ. Res. Public. Health 2022, 19, 7446, doi:10.3390/ijerph19127446.

Menesini, E.; Salmivalli, C. Bullying in Schools: The State of Knowledge and Effective Interventions. Psychol. Health Med. 2017, 22, 240–253, doi:10.1080/13548506.2017.1279740.

Hu, J.C.-C.; Simmer, J.P. Developmental Biology and Genetics of Dental Malformations. Orthod. Craniofac. Res. 2007, 10, 45–52, doi:10.1111/j.1601-6343.2007.00384.x.

de La Dure-Molla, M.; Fournier, B.P.; Manzanares, M.C.; Acevedo, A.C.; Hennekam, R.C.; Friedlander, L.; Boy-Lefèvre, M.-L.; Kerner, S.; Toupenay, S.; Garrec, P.; et al. Elements of Morphology: Standard Terminology for the Teeth and Classifying Genetic Dental Disorders. Am. J. Med. Genet. A. 2019, 179, 1913–1981, doi:10.1002/ajmg.a.61316.

Atik, E.; Önde, M.M.; Domnori, S.; Tutar, S.; YiÄŸit, O.C. A Comparison of Self-Esteem and Social Appearance Anxiety Levels of Individuals with Different Types of Malocclusions. Acta Odontol. Scand. 2021, 79, 89–95, doi:10.1080/00016357.2020.1788720.

Akpasa, I.O.; Yemitan, T.A.; Ogunbanjo, B.O.; Oyapero, A. Impact of Severity of Malocclusion and Self-Perceived Smile and Dental Aesthetics on Self-Esteem among Adolescents. J. World Fed. Orthod. 2022, 11, 120–124, doi:10.1016/j.ejwf.2022.05.001.

Mullan, V.M.R.; Golm, D.; Juhl, J.; Sajid, S.; Brandt, V. The Relationship between Peer Victimisation, Self-Esteem, and Internalizing Symptoms in Adolescents: A Systematic Review and Meta-Analysis. PloS One 2023, 18, e0282224, doi:10.1371/journal.pone.0282224.

Skrzypiec, G.; Alinsug, E.; Nasiruddin, U.A.; Andreou, E.; Brighi, A.; Didaskalou, E.; Guarini, A.; Kang, S.-W.; Kaur, K.; Kwon, S.; et al. Self-Reported Harm of Adolescent Peer Aggression in Three World Regions. Child Abuse Negl. 2018, 85, 101–117, doi:10.1016/j.chiabu.2018.07.030.

Reviewer 4 Report

Before I begin my review, I would like to highlight that the manuscript, in its current form, exhibits significant methodological and terminological flaws that require immediate attention. These issues must be addressed as they impact the overall quality and reliability of the study. Therefore, I strongly recommend a rejection of the manuscript to rectify these concerns and enhance its scientific rigor.

The title. In my opinion, the use of the term "abnormal dental features" in the title is overly strong. I would recommend considering a more neutral term to describe the dental characteristics under investigation.

The abstract. The statement in the abstract, "As oral features can promote psychological distress, also found in bullies, we aimed to study the association between orofacial conditions and bullying," suggests that the authors believe there is a potential relationship between certain oral features or conditions and the psychological distress experienced by both individuals who are bullied and individuals who engage in bullying behavior. Is that what authors' intention was?

As a dental clinician, I have carefully evaluated the manuscript and unfortunately, I must recommend rejection with a sense of sorrow. Throughout the manuscript, I have identified several flaws, including inadequate referencing and unsupported conclusions. As dental professionals, it is crucial for us to rely on robust evidence and accurate referencing to ensure the validity and reliability of research findings. Furthermore, the unsupported conclusions drawn in the manuscript raise concerns about the reliability of the study's outcomes. I encourage the authors to address these methodological and referencing issues comprehensively, providing stronger evidence and more robust support for their conclusions. By doing so, the manuscript would significantly improve and contribute to the dental literature.

Author Response

We thank the reviewer for his comments.

As requested by the reviewer, the title has been corrected as followed: Association between dentofacial features and bullying from childhood to adulthood : a systematic review

The term “abnormal” has been removed to remain neutral towards these patients.

The reviewer is completely right and this point (Bullies/bullied) is now corrected in the new version of the manuscript: “bullies” has been replaced by “bullied”.

Authors have fully revised references and added new references to provide stronger evidence:

De Ridder, L.; Aleksieva, A.; Willems, G.; Declerck, D.; Cadenas de Llano-Pérula, M. Prevalence of Orthodontic Malocclusions in Healthy Children and Adolescents: A Systematic Review. Int. J. Environ. Res. Public. Health 2022, 19, 7446, doi:10.3390/ijerph19127446.

Menesini, E.; Salmivalli, C. Bullying in Schools: The State of Knowledge and Effective Interventions. Psychol. Health Med. 2017, 22, 240–253, doi:10.1080/13548506.2017.1279740.

Hu, J.C.-C.; Simmer, J.P. Developmental Biology and Genetics of Dental Malformations. Orthod. Craniofac. Res. 2007, 10, 45–52, doi:10.1111/j.1601-6343.2007.00384.x.

de La Dure-Molla, M.; Fournier, B.P.; Manzanares, M.C.; Acevedo, A.C.; Hennekam, R.C.; Friedlander, L.; Boy-Lefèvre, M.-L.; Kerner, S.; Toupenay, S.; Garrec, P.; et al. Elements of Morphology: Standard Terminology for the Teeth and Classifying Genetic Dental Disorders. Am. J. Med. Genet. A. 2019, 179, 1913–1981, doi:10.1002/ajmg.a.61316.

Atik, E.; Önde, M.M.; Domnori, S.; Tutar, S.; YiÄŸit, O.C. A Comparison of Self-Esteem and Social Appearance Anxiety Levels of Individuals with Different Types of Malocclusions. Acta Odontol. Scand. 2021, 79, 89–95, doi:10.1080/00016357.2020.1788720.

Akpasa, I.O.; Yemitan, T.A.; Ogunbanjo, B.O.; Oyapero, A. Impact of Severity of Malocclusion and Self-Perceived Smile and Dental Aesthetics on Self-Esteem among Adolescents. J. World Fed. Orthod. 2022, 11, 120–124, doi:10.1016/j.ejwf.2022.05.001.

Mullan, V.M.R.; Golm, D.; Juhl, J.; Sajid, S.; Brandt, V. The Relationship between Peer Victimisation, Self-Esteem, and Internalizing Symptoms in Adolescents: A Systematic Review and Meta-Analysis. PloS One 2023, 18, e0282224, doi:10.1371/journal.pone.0282224.

Skrzypiec, G.; Alinsug, E.; Nasiruddin, U.A.; Andreou, E.; Brighi, A.; Didaskalou, E.; Guarini, A.; Kang, S.-W.; Kaur, K.; Kwon, S.; et al. Self-Reported Harm of Adolescent Peer Aggression in Three World Regions. Child Abuse Negl. 2018, 85, 101–117, doi:10.1016/j.chiabu.2018.07.030.

Round 2

Reviewer 2 Report

its well organized 

Reviewer 4 Report

Dear authors,

First and foremost, I would like to express my satisfaction with receiving this paper for re-evaluation. I am grateful, primarily to the editor, for making a fair assessment and not automatically rejecting the paper after my negative review. In the new version of the paper, all of my concerns have been adequately addressed.

As a clinical dentist, my focus was primarily on the accuracy from a clinical dental perspective, which is not actually the most important topic nor the greatest scientific contribution of this paper. I momentarily overlooked the importance of this subject, its relevance, and the scientific contribution in an area that is vastly underexplored. I am pleased to recommend accepting the paper in its current version.

After such an injustice I have caused, the least I can do is sign my name.

Respectfully, Bojan Petrović Faculty of Medicine, Department of Dental Medicine, University of Novi Sad, Serbia